# Involvement of Il-33 in the Pathogenesis and Prognosis of Major Respiratory Viral Infections: Future Perspectives for Personalized Therapy

**DOI:** 10.3390/biomedicines10030715

**Published:** 2022-03-19

**Authors:** Giuseppe Murdaca, Francesca Paladin, Alessandro Tonacci, Matteo Borro, Monica Greco, Alessandra Gerosa, Stefania Isola, Alessandro Allegra, Sebastiano Gangemi

**Affiliations:** 1Department of Internal Medicine, Ospedale Policlinico San Martino, 16132 Genoa, Italy; puell-a@hotmail.it (F.P.); a.gerosa92@gmail.com (A.G.); 2Clinical Physiology Institute, National Research Council of Italy (IFC-CNR), 56124 Pisa, Italy; atonacci@ifc.cnr.it; 3Internal Medicine Department, San Paolo Hospital, 17100 Savona, Italy; borromatteo@libero.it (M.B.); monicagreco89@gmail.com (M.G.); 4Department of Clinical and Experimental Medicine, School and Operative Unit of Allergy and Clinical Immunology, University of Messina, 98125 Messina, Italy; stefania.isola@unime.it (S.I.); gangemis@unime.it (S.G.); 5Department of Human Pathology in Adulthood and Childhood “Gaetano Barresi”, Division of Hematology, University of Messina, 98125 Messina, Italy; alessandro.allegra@unime.it

**Keywords:** IL-33, chronic lung diseases, respiratory viral infection, SARS-CoV-2, biological drugs

## Abstract

Interleukin (IL)-33 is a key cytokine involved in type-2 immunity and allergic airway disease. At the level of lung epithelial cells, where it is clearly expressed, IL-33 plays an important role in both innate and adaptive immune responses in mucosal organs. It has been widely demonstrated that in the course of respiratory virus infections, the release of IL-33 increases, with consequent pro-inflammatory effects and consequent exacerbation of the clinical symptoms of chronic respiratory diseases. In our work, we analyzed the pathogenetic and prognostic involvement of IL-33 during the main respiratory viral infections, with particular interest in the recent SARS-CoV-2virus pandemic and the aim of determining a possible connection point on which to act with a targeted therapy that is able to improve the clinical outcome of patients.

## 1. Introduction

In December 2019, an outbreak caused by Coronavirus disease 2019 (COVID-19) occurred in Wuhan, Hubei province, China [1]. SARS-CoV-2 is a positive single-stranded enveloped RNA virus (ssRNA) that belongs to the beta-coronavirus genus of the Coronaviridae family [2]. As with other infections caused by respiratory viruses, a process of recognition by cytotoxic T lymphocytes also occurs following SARS-CoV-2 infection, resulting in the overproduction of a wide range of proinflammatory mediators, known as a “cytokine storm” (CS), an uncontrolled production of soluble inflammatory markers that, in turn, support an aberrant systemic inflammatory response, which seems to be the main cause of the clinical severity of respiratory infection, up to acute respiratory distress syndrome (ARDS). Mast cells and basophils also appear to be involved in the genesis of CS; their activation, in fact, would be able to determine the procoagulant state characteristic of this type of infection [2,3]. In this context, the alarmin known as interleukin-33 (IL-33), traditionally involved in allergic and autoimmune diseases (such as rheumatoid arthritis, systemic sclerosis, atopic dermatitis and food allergies), assumes a role of particular interest, as it is involved in pro-inflammatory processes [4]. IL-33 acts by binding to a receptor complex known as the accessory protein of the ST2 and IL-1 receptor, expressed on a series of innate and adaptive immune cells, to initiate inflammatory pathways dependent on the myeloid differentiation factor 88 (MyD88) response and exerting pleiotropic effects [5]. The soluble form of ST2, as a down-regulation mechanism of IL-33, has been identified as a reliable biomarker of poor prognosis in cardiovascular disease, acute respiratory distress syndrome (ARDS) and other inflammatory conditions. For example, in children with acute lower respiratory infection (ALRI), local activation of the IL-33/ST2 axis has been shown to be associated with a more severe disease evolution, with the need for ventilatory support [6]. More recently, a link has been discovered between IL-33 and lung damage in pulmonary viral infections and chronic lung diseases. In fact, IL-33 is known to increase airway inflammation, mucus secretion and Th2 cytokine synthesis in the lungs, following respiratory infections such as influenza virus, RSV and rhinovirus infection [7]. However, IL-33 can also stimulate the activity of antiviral cytotoxic T cells and the production and release of antibodies [7]. Similarly to other respiratory virus infections, exposure to the SARS-CoV-2 peptide also causes IL-33 expression, correlating with T cell activation and lung disease severity. Although the increase in IL-33 levels is considered a predictor of severe COVID-19, its precise role at different stages of the disease is still unclear. However, the activation role of IL-33 on antiviral CD8 T cells, with consequent possible elimination of the virus in long-lasting infections, is known. As a result, targeting the IL-33-ST2 axis using monoclonal antibodies could prove to be an effective strategy for controlling the COVID-19 pandemic [8,9,10].

In this review, we wanted to collect and analyze the current knowledge regarding the role of IL-33 in the pathogenesis and prognosis of respiratory viral infections, with particular interest in SARS-CoV-2 virus, with the aim of determining a possible connection point on which to act with a targeted therapy, in order to improve the clinical outcome of patients. In Table 1, we have reported the articles analyzed in our review.

## 2. IL-33: Synthesis, Receptor Effects and Pathogenetic Role in Respiratory Diseases

IL-33 is a nuclear cytokine derived from tissues of the IL-1 family. It is found widely expressed in various cells of the human body, including endothelial, epithelial and fibroblast-like cells. Its function is that of an alarm signal (alarmin), released in the event of cell or tissue damage to alert the immune cells that express the ST2 receptor (IL-1RL1) [11]. IL-33 is a dual-function cytokine based on the size of its structure: the full-length IL-33 protein (flIL-33) acts as a regulator of the intranuclear gene, while mature IL-33 (mIL-33) is released as a result of cell damage or necrosis. Both forms, flIL-33 and mIL-33, can bind and signal through ST2, which is predominantly expressed by immune cells involved in innate immunity, including mast cells, ILC2, macrophages, basophils, natural killer cells (NK cells), dendritic cells (DC), and eosinophils. Furthermore, ST2 is expressed by cells that participate in adaptive immunity, such as CD4+, CD8+ T cells, and T-regulatory cells (Tregs) [12]. By binding to ST2, IL-33 activates target cells and stimulates them to secrete cytokines and growth factors that promote and/or regulate local and systemic immunity [13]. Once bound to the ST2 receptor, IL-33 intensely stimulates the production of Th2-associated proinflammatory cytokines, including IL-4, IL-5, and IL-13 [14]. Regarding chronic respiratory diseases, IL-33 concentrations have been shown to be significantly elevated in the sputum and bronchoalveolar lavage fluid (BALF) of patients with bronchial asthma. In these patients, IL-33 causes a wide range of effects, including neutrophil migration, mast cell activation, osteoclast and osteoblast function, wound healing, and others [15].

## 3. Role of IL-33 in Influenza A and B Virus Infection

In a research study conducted both in vivo and in vitro, Ronan Le Goffic et al. evaluated both the expression of IL-33 in H1N1 influenza A virus–induced pulmonary infection in mice after 3 days from infection and the induction of IL-33 in response to infections with different strains of the same virus in cultured human and murine epithelial cells. The authors found a significant (*p* < 0.001) relative and absolute increase in IL-33 mRNA expression in the virus-infected group, compared with controls. Moreover, a significant correlation was found between IL-33 mRNA induction and mRNA levels of TNF-a, IFN-g, and IL-6, but not with IL-1b, thus suggesting IL-33 is a member of the classic pro-inflammatory cytokines. The evaluation of IL-33 protein showed, interestingly, an increase on day 3 after infection as compared with non-infected lungs, thus suggesting an inducible behavior of the cytokine. A similar result was found in the bronchoalveolar lavage of virus-infected mice, compared with the control group. By analyzing the response of epithelial cells, the authors showed that the transformed murine respiratory epithelial cell line, MLE-15, presented a threefold increase in IL-33 mRNA expression after infection with influenza A virus compared with non-infected cells. Moreover, the increasing trend was found at 20 h after infection and reached the maximum levels at 40 h. By infecting human pulmonary epithelial cell line A549 (type II pneumocyte cell line) with different strains of influenza virus, the authors showed similar results with a significant increase (2–4 fold) in IL-33 at transcript levels. These results evidenced that the expression of IL-33 is induced by influenza A virus infection, both at the mRNA and protein level, thus supporting its role as alarmin cytokine [16]. Jia-Rong Bian et al. evaluated serum levels of pro-inflammatory cytokines in adult patients with seasonal influenza infection. They enrolled 72 patients with a laboratory-confirmed seasonal influenza infection diagnosis and 30 healthy controls. Among the infected patients, 24 presented with influenza A and 48 with influenza B. Their lab results showed higher serum concentrations of IL-6, IL-33, and TNF-α at admission, compared to controls. Higher levels of IL-33 were found in the influenza A-infected patients than in the influenza B-infected patients. In the influenza A-infected patients, evaluation at day 6 showed a significant reduction in IL-6, IL-33, and TNF-α compared to their at admission [17]. These results are supported by previous research performed by Kayamuro H et al. The authors analyzed mice intranasally immunized with recombinant influenza virus hemagglutinin co-administered with various cytokines as mucosal vaccine adjuvants. Of all cytokines analyzed, the IL-1 family of cytokines, which comprehend IL-33, presented the highest power to induce the production of specific IgGs, both IgG1 and IgG2a, and soluble IgA in the mucosal system. Moreover, the authors found the highest levels of IL-4 and IL-5 in splenocytes from immunized mice with hemagglutinin plus IL-18 or IL-33, thus suggesting that IL-1 family cytokines may be able to elicit both Th1- and Th2-type cytokine responses. Analyzing the cell response, the authors found that IL-18 and IL-33 were able to induce high-avidity CD8+ cytotoxic lymphocytes, thus sustaining those cytokines as useful adjuvants of effective mucosal influenza vaccines [50]. According to these findings, Xi-zhi J. Guo et al. demonstrated that production of IL-33, induced by IL-17A secreted from the responding γδ T cells, was increased in the mucosa after infection of a mouse model by influenza virus. This mechanism subsequently led to a local Type-2 immune response with increased accumulation of Type-2 innate lymphoid cells and T regulatory cells in the lung, thus promoting tissue repair and lung integrity following infection and contributing to protection against mortality. Moreover, authors observed a correlation between IL-17A, IL-33, and amphiregulin in nasal washes of human influenza-infected infants. Their findings supported the role of IL-33, especially in tissue recovery, suggesting a possible therapeutic target [51]. A further support of IL-33 involvement in protection and restoration after influenza infection has been demonstrated by Monticelli et al. In mice, authors showed the presence of a particular innate lymphoid cell expressing CD90, CD25, CD127 and T1-ST2 on the surface that promotes lung-tissue homeostasis after infection with influenza virus. Through the administration of anti-IL-33 mAb, they tested whether the IL-33 signaling was necessary for the maintenance of airway epithelial integrity. Blockade of IL-33 signaling pathway resulted in a significant decrease in both frequency and total cell numbers of innate lymphoid cells in the lungs compared to those in controls. Moreover, they found that the blockade of IL-33 signaling was associated with severely impaired lung function, desaturation, and the presence of epithelial cell necrosis and bronchial degeneration within the airways [19]. Further supporting results were presented by Robinson et al. They evaluated the impact of IL-33 in mice infected by influenza virus and superinfected by Staphylococcus aureus. Their results showed that a reduction in IL-33 during superinfection correlated with a negative outcome, in contrast with what was seen during normal Influenza infection, and, more importantly, an improved bacterial clearance after restoration of IL-33. Interestingly, they demonstrated that the improvement related to IL-33 restoration was not due to innate lymphoid cells or type-2 macrophage or related to an increase in pro-inflammatory cytokines IL-6 or TNF-α but was related to an increase in neutrophil count and neutrophil activity [20]. Another study, performed by Chae Won Kim et al., showed that exogenous administration of IL-33 was related to a better outcome in mice infected with influenza virus and an enhanced protective effect of antiviral immunity. Intranasal administration of IL-33 was shown to increase, in lung, the numbers of innate lymphoid cells, eosinophils, and dendritic cells that, in turn, are able to activate CD8+ T-cell responses against influenza infection. Moreover, exogenous IL-33 enhanced the secretion of IL-12 in bronchoalveolar lavage fluids post-infection, amplified IL-12 production, and increased the maturation of dendritic cells. Interestingly, only exogenous IL-33 was able to increase the number of IFN-γ-producing CD8+ T-cells. Taking their results together, the authors showed that exogenous IL-33 exhibits an antiviral effect in the lung against influenza infection through a reinforced antiviral immunity, resulting in increased survival and viral clearance [21].

Based on these findings, in influenza infection, IL-33 seems to act as a protective factor, inducing better viral clearance and promoting tissue repair and lung integrity. Evidence showed that this mechanism may be explained through an increase in antigen-presenting cells, such as macrophages and dendritic cells, that can induce improved CD9+-T-cell activation. The rapid behavior presented by IL-33 and IL-33 mRNA supports this hypothesis and confirms its role as alarmin and pro-inflammatory cytokine. In vitro study confirmed that blockade of IL-33 is correlated with a worse outcome after influenza infection and, moreover, that a restoration of its levels is linked with an improvement of final outcome. Further research is needed in order to improve our understanding and develop a possible future treatment for patients diagnosed with influenza virus infection.

## 4. Role of IL-33 in Rhinovirus Infection

Rhinovirus (RV) is a positive-sense single-stranded RNA virus, classified into rhinovirus A and rhinovirus B based on differential susceptibility to capsid-binding compounds, both subtypes that use ICAM-1 as cellular entry receptor [22]. In non-asthmatic individuals, symptoms of infection are usually confined to the upper respiratory tract, whereas in patients suffering from asthma or other respiratory diseases, symptoms affecting the lower respiratory tract are more frequent, such as cough, shortness of breath, chest tightness and wheezing [52]. RV-induced exacerbations in asthmatics have been linked to the defective expression of protective interferons of type I and III in epithelial cells, which are also responsible for the production of various inflammatory mediators, responsible for Th2-based stimulation and, therefore, susceptibility to development of acute allergic inflammation of the airways [23,25,27]. Epithelium-derived “innate cytokines” include IL-25 (or IL-17E), IL-33, and thymic stromal lymphopoietin (TSLP), which are produced in response to RV infection and play a role in the maturation of Th2 cells through activation of dendritic cells [24,26]. In this regard, Jurak LM et al. [28], demonstrated that in asthmatic subjects, IL-33 stimulates type-2 inflammatory responses to RV, but has little effect on type-1 immune responses, probably as a consequence of the differential regulation of the IL-33 receptor with increased expression of the ST2 form in asthmatic cells. Furthermore, the same study also showed how RV and IL-33 act together to increase IFNγ production by NK cells. Several studies conducted in vitro on bronchial epithelial cells and in vivo on mouse models have shown how the production of IL-33 following RV infection is determined by the activation of two specific signaling pathways: that mediated by toll-like receptors 3 (TLR3) and 2 (TLR2). These are in turn responsible for the secretion of specific chemokines such as CXCL-10, resulting in the recruitment of macrophages into the lung and the development of inflammation [29,32,53,54]. In asthmatics, IL-33 would also be able to facilitate viral capture by the pulmonary vascular endothelium by the expression of ICAM-1, thus intensifying the inflammation orchestrated by the endothelium (from IL-1β, IL-6), promoting the growth (from G-CAS, GM CSF) and recruitment of mostly innate immune cells: neutrophils and eosinophils [31]. In consideration of the role that IL-33 plays in the severe asthma exacerbations evoked by rhinovirus, this makes it one of the potential targets of anti-IL-33 monoclonal antibody biotherapy. Blocking IL-33, in fact, attenuates respiratory inflammation in all phases of the infection, suppressing type-2 inflammation, restoring antiviral immunity, and increasing viral clearance [18,39,40].

## 5. Role of IL-33 in Respiratory Syncytial Virus Infection

A recent review of the literature evaluated the impact of type-2 response against respiratory syncytial virus (RSV), aimed at finding a possible explanation for the higher airway hyperreactivity evidenced after RSV infection [55]. In nasal aspirates of hospitalized infants with RSV infection, increased amounts of IL-33 together with IL-13 were found [38]. This finding is supported by a previous study performed by Bertrand et al. in children with RSV bronchiolitis; the authors found a high level of IL-33 in nasopharyngeal aspirates of patients with a family history of atopy [35]. In line with these findings, García-García et al. measured IL-33 levels from nasopharyngeal aspirates in children infected with RSV and found an association with bronchiolitis, especially when co-infection with other respiratory virus occurred [56]. These findings were then confirmed by Vu et al. in a study of infants hospitalized for RSV infection; the authors demonstrated that levels of IL-4, IL-13, IL-33, and IL-1β were significantly higher in nasal aspirates of patients with severe disease compared with those of patients with moderate disease [33,36]. A mouse model of RSV infection showed an increase in the production of IL-33 together with elevated expression of its receptor ST2, followed by a massive infiltration of CD45+ST2+ cells in the lungs. Blocking ST2 with an anti-ST2 monoclonal antibody reduced both the RSV-induced eosinophil recruitment and the amount of Th2-associated cytokines, especially IL-13 and Th17-type cytokine IL-17A in the lungs of RSV-infected mice [34]. To further support this evidence, a mouse model of RSV infection showed that macrophages and especially dendritic cells in lungs are a source of IL-33 [35], confirmed by mRNA measurement [37]. Indeed, epithelial cells also increase the expression of IL-33 after RSV infection [30]. The increase in IL-33 expression has been shown to be mediated by interaction with toll-like receptors 3 and 7 [37]. Furthermore, Feifei Qi et al., in another study, demonstrated that RSV infection is able to enhance both the expression of mRNA for MAPK molecules, including p38, c-Jun N-terminal kinase (JNK) 1/2, and extracellular signal-regulated kinase (ERK) 1/2, and the levels of mitogen-activated protein kinase (MAPK) in lung macrophages, which in turn can induce the production of IL-33 [57]. Type-2 innate lymphoid cells represent one of the targets of IL-33 and are activated after RSV infection [58]. IL-33 production and the number of type-2 innate lymphoid cells differ greatly after RSV infection based on age: neonatal mice produce more IL-13 and IL-33 in response to infection than adult mice, as well as the number of type-2 innate lymphoid cells [59]. In agreement with this result, Stier MT et al. showed that in RSV-infected IL-33 knock-out mice, there was a significant decrease in the total lung concentration of IL-13, thus suggesting an important role for IL-33 in ILC2 activation in a murine model of RSV infection [60]. Xu Han et al. [61] showed the presence of an innate immune lymphocyte-like population of “natural helper” cells that plays a role after RSV infection. This subcellular population has been linked to the increased production of Th2-associated cytokines, such as IL-4, IL-5, and IL-10 after RSV infection. Surprisingly, this was reduced by blockage of IL-33 with antibody, thus suggesting that IL-33 is necessary for activating Th2-type natural helper cells. In a mouse model, Yi-Hsiu Wu et al. [7] demonstrated that IL-33 is crucial for the activation of ILC2s and the development of airway hyperreactivity and airway inflammation. Specifically, they showed that IL-33^-/-^ mice developed lower airway hyperreactivity, neutrophil infiltration, and IL-5 and IL-13 production. Moreover, during RSV infection, the authors showed an increased number of type-2 innate lymphoid cells in lungs with an increased release of IL-5 and IL-13, induced by IL-33; in fact, IL-33^−/−^ mice did not show the same characteristics. Further results showed that RSV infection through IL-33 produced by lung myeloid cells, such as alveolar macrophages, interstitial macrophages, and dendritic cells, may act in cellular recruitment but not in airway hyperreactivity. Recent research performed by Liwen Zhang et al. [38] evaluated the role of the NF-κB/IL-33/ST2 axis in RSV-induced acute bronchiolitis both in mice and humans. The authors found that IL-33 concentrations in infants were significantly increased after RSV infection compared to normal subjects. Moreover, to further evaluate the role of IL-33 in inducing the Th2 environment after RSV infection, in mice, they showed an increased level of IL-33 as well as of its receptor ST2, p65, and macrophages and eosinophils in the bronchoalveolar lavage. Subsequently, the addition of an anti-IL-33 antibody produced a reduction of mRNA expression of Th2 cytokines IL-4, and IL-10 and of eosinophils and macrophages in lung tissues. The importance of IL-33 in inducing a pro-inflammatory environment has been further supported by Carolina Augusta Arantes Portugal et al. [6]. They evaluated the IL-33/ST2 axis in acute lower respiratory infection in 73 children of less than 5 years of age. Their results showed an early and more prominent local activation of the IL-33/ST2 axis in children with more severe RSV infection who needed ventilator support. In fact, IL-33 levels in the nasopharyngeal aspirate on the first day following admission were associated with a higher risk for mechanical ventilation. Results in line with those previous mentioned were also reported by Jing Liu et al. in 2015 [58]. The authors showed that RSV infection correlated with an increase in the number of IL-13-producing natural helper cells as well as the expression of IL-13 mRNA in natural helper cells. Moreover, they showed that IL-13 production was induced by IL-33 release, which was markedly increased in the lungs of mice after RSV infection, both in vivo and in vitro. The addition of an anti-IL-33 antibody was correlated with a reduction in IL-13 production. These findings are in line with previous results shown by Jordy Saravia et al. [59], in which concentrations of IL-33 and IL-13 were significantly elevated at hospitalization compared to at 4 weeks.

In conclusion, all of these findings seem to support the correlation between RSV infection and an increase in IL-33 release that, in turn, is crucial for induction of atopy, airway hypersensitivity, and worse-outcome bronchiolitis. Mouse models of RSV infection showed that administration of an anti-IL-33-receptor antibody that blocks the IL-33 cascade can reduce both the Th2 and Th17 environment (especially IL-13 and IL-17A) and the eosinophil recruitment induced by IL-33 after infection. These findings are promising, and further research is needed to better understand the mechanisms and speculate about a possible target preventive therapy.

## 6. Role of IL-33 in Human Adenovirus Infection

Human adenoviruses (HAdVs) are double-stranded nonenveloped DNA viruses that are commonly associated with a wide variety of respiratory, ocular, and gastrointestinal diseases. Generally well-tolerated, in some high-risk patients (e.g., organ transplant recipients, patients with HIV infection or congenital immunodeficiency syndromes) they may develop a more severe disease that may lead to death [41,62]. HAdVs are classified into seven species, ranging from A through G, and there are currently more than 50 serotypes, each associated with different clinical manifestations of infection [42,43]. AdV infections may occur in healthy children or adults in closed or crowded settings, such as daycare or in hospitals, and most epidemics occur in the winter or early spring. Transmission can occur via aerosolized droplets, direct conjunctival inoculation, orofecal spread, and exposure to infected tissue or blood or environmental surfaces (linen, pillows, lockers) [62]. The incubation period ranges from 2 to 14 days and depends upon viral serotype and mechanism of transmission [41]. Adenoviruses are strongly immunogenic, and this has consequences not only for infection outcomes and prevention, but also for the use of AdVs as vectors for gene therapy, vaccines, or cancer gene therapy [63]. Following the first HAdV infection, the first immune response is made by a subset of B cells that innately secretes immunoglobulin M (IgM), even without prior antigen exposure. In the weeks following the infection, the adaptive immune system produces specific anti-AdV IgG antibodies against three exposed accessible proteins: hexon, fiber, and penton. The entry of the virus into nonimmune cells is initiated through binding of the fiber knobs to entry, while entry pathways into immune cells (macrophages and dendritic cells) may be different and could be modulated by cytokines or antibodies. For example, the entry of HAdV-C2/5 into polarized airway epithelial cells from the apical side is enhanced by cytokines and chemokines, including interleukin 8 and TNF alpha [64]. HAdV hexon-specific CD4^+^ T and CD8^+^ T (less common) cells have been found among peripheral blood lymphocytes of almost all individuals of all ages, and there is evidence that cytotoxic T lymphocytes (CTLs) specific to hexon are protective against infections [63]. Adenoviral transduction also stimulates IL-33 expression in endothelial cells in a manner that depends on the DNA-binding protein MRE11 (a sensor of DNA damage that can also be activated by adenoviral DNA) and the antiviral factor IRF1, an essential transcriptional regulator of the DNA damage response [65]. As previously mentioned, adenoviruses are also used as vectors for gene therapy and for vaccines. The significant advantages of AdV vectors over other vector systems include efficient transduction of a variety of cell types, both quiescent and dividing, making them optimal for certain applications. One of the main challenges is the AdV-induced toxicity, which occurs within minutes in the absence of viral gene expression and is attributed to the innate response. As the vector distributes in the blood, it interacts with cells of the reticuloendothelial system (RES) and induces the release and/or production of several proinflammatory cytokines including IL-6, TNFa, IL-8, GM-CSF and MIP-2 [8]. As for interleukin 33, in their study, Yin et al. constructed a recombinant replication-deficient adenovirus encoding soluble ST2-human immunoglobulin (Ig) G1 Fc (sST2-Fc), and studied its beneficial effect in a murine model of ovalbumin (OVA)-sensitized asthma, reporting that the protective effect of sST2-Fc in allergic lung inflammation is related to blocking IL-33/ST2L signaling. The T2 gene is a member of the IL-1 receptor family; soluble ST2 is known to be implicated in a variety of immune disorders in humans. IL-33 is a specific ligand of ST2L, and IL-33/ST2L signaling appears to play a crucial role in Th2-type immunopathological changes [66]. Luzina et al. used a replication-deficient adenoviral (Ad5) carrying two mycobacterial antigens, Ag85A and Mtb32, in an OVA-induced asthmatic mouse model to investigate the protective effects of Ad5-gsgAM against allergic asthma. Ad5-gsgAM-immunized mice showed decreased serum IL-33 as compared to Ad5. Exogenous IL-33 abrogated the protective effects of Ad5-gsgAM, revealing that suppression of the IL-33/ST2 axis substantially contributed to protection against allergic inflammation, Both Ad5-gsgAM and BCG decreased serum IL-33 as compared to Ad5 [67].

Even if adenoviruses are known to be strongly immunogenic, there are just a few studies that correlate these viruses with the production of IL-33 in the human body. More information has been obtained using adenoviruses as vectors.

## 7. Role of IL-33 in Human Metapneumovirus Infection

Human metapneumovirus (hMPV) is a clinically relevant single-stranded RNA virus that belongs to the genus Metapneumovirus in the new virus family Pneumoviridae [68]. Since its discovery in 2001, hMPV has been increasingly recognized as responsible for respiratory infections, especially in infants, children, elderly, and immune-compromised subjects [69]. HMPV causes a different variety of disease manifestations, from upper respiratory tract infection to severe lower respiratory tract infections (e.g., bronchiolitis and pneumonia) [70]. The viral envelope contains three transmembrane surface glycoproteins: the highly glycosylated attachment glycoprotein G, the fusion protein F, and the small hydrophobic protein SH. Proteins F and G promote viral binding and penetration into the host cell cytoplasm, thus leading to infection [45]. Protein G has also been shown to inhibit the IFN-I response, as well as to promote the recruitment of neutrophils into the alveolar space in the lungs of mice infected with hMPV [49]. hMPV preferentially infects airway epithelial cells (AECs) and causes the latter to produce some key mediators in the development of the pulmonary inflammatory process, such as thymic stromal lymphopoietin (TSLP), a cytokine similar to IL-7 derived from epithelial cells, capable of promoting inflammation induced by both type-1 (TH1) and type-2 (allergic-TH2) profiles and the cytokine IL-33 [46]. Recent studies have highlighted how the secretion of IL-33 has important implications for the exacerbation of asthma induced by viral infection, hypothesizing a mechanism through which respiratory viruses, which are the classic Th1 triggers, could promote the type-2 inflammation in susceptible individuals [56]. TLSP and IL-33 activate dendritic cells (DCs) by inducing the expression of OX40L, which stimulates naive CD4+ T cells to differentiate into Th2 cells. DCs activated by TLSP in turn produce CCL17, a Th2-enhancing chemokine and natural killer (NK) cell attractor. By creating a microenvironment capable of stimulating Th2-mediated responses, hMPV could hinder or delay a more efficient Th1 antiviral response, thus aggravating the clinical manifestations of the viral infection [44].

In the context of allergic inflammation, IL-33 promotes a number of pro-inflammatory factors, such as eosinophil survival and the expression of the intercellular adhesion molecule (ICAM)-1. Furthermore, IL-33 stimulates the significant release of the pro-inflammatory cytokine IL-6 and chemokines CXCL8 and CCL2 from eosinophils [71]. The release of IL-33 is also responsible for the remodeling processes that occur at the bronchial level already in infancy, thus predisposing the subject to a greater clinical severity of the disease.

## 8. Role of IL-33 in SARS-CoV-2 Infection

The outbreak of SARS-CoV-2, which first emerged in China in December 2019, spread rapidly all over the world, and although vaccine programs are in progress, it is still causing many deaths worldwide. From the beginning, it was evident that the main target of the COVID-19 infection was the respiratory system, although it is now known that it is a multi-organ disease characterized by a hyperinflammatory status. Despite the remarkable discoveries regarding the pathophysiology of the infection, the complexity of the mechanisms involved in COVID-19 make this disease still unclear. Moreover, the involvement of many mediators including several inflammatory cytokines make investigations even more complex. Among them, IL-33 was extensively studied, and its role is still debated. IL-33 belongs to the IL-1 family and acts as an alarmin that reveals cellular damage or infection. To act, this molecule needs to be cleaved by proteases to bind ST2 receptor (also known as IL-1RL1) and finally activate the NF-κB pathway in various innate and adaptive immune cells. It was also noticed that IL-33 receptor exists in three isoforms: full-length transmembrane form (ST2L), membrane-bound variant (ST2V), and secreted soluble form (soluble ST2, sST2). The latter, as discussed below, proved to have a notable role upon COVID-19 infection. To date, several studies have demonstrated a significant correlation between SARS-CoV-2 infection and IL-33. More specifically, it was speculated that IL-33 levels correlate with disease severity. Indeed, in their observational study, Burke H. et al. [47], using a multiplex cytokine assay, analyzed serum concentrations of several pro-inflammatory cytokines including IL-33 in 100 hospitalized patients with COVID-19. They found that increased levels of IL-33 were associated with adverse outcome. Hence, Markovic S. et al. [48] conducted a study on 220 COVID-19 patients, focusing on disease severity. They divided patients among three disease groups and showed that IL-33 levels were higher among patients in the worst clinical group. Moreover, the same positive correlation was noticed regarding radiological severity, as IL-33 levels were higher among patients presenting with pulmonary multifocal consolidation and acute respiratory distress syndrome (ARDS). Similar observations were further confirmed by Liang Y. et al. [10], who conducted an in vitro study on two human lines of epithelial cells, Fadu and LS513, to evaluate whether SARS-CoV-2 infection induces IL-33 expression. They demonstrated that SARS-CoV-2 infection stimulates IL-33 expression in human epithelial cells as a damage consequence; indeed, IL-33 transcript levels increased in both cell lines 72 h after infection. Additionally, Munitz A. et al. [72] noticed a significant elevation of IL-33 in moderate and severe patients compared with mild ones. Furthermore, focusing on seroconversion, they found that IL-33 clustered together with anti-receptor binding domain (RBD) IgG in the moderate and severe cohorts. This observation was also confirmed by Stanczak M.A. et al. [73]. In fact, in their study they discovered that SARS-CoV-2 peptide induced production of IL-33, clustered together with IgG titers. Surprisingly, IL-33 production was more closely related to IgG titers than any other parameter of the study, including subject age. Moreover, researchers noticed that after recovery from COVID-19, individuals still had persisting, circulating PBMCs that produced IL-33 in response to virus-specific T-cell activation, which correlated with seropositivity. Interesting results were also noted by Zeng G. et al. [9]. Hence, they focused on soluble ST2 levels among COVID-19 patients and their relationship with inflammatory status and disease severity. Firstly, a positive correlation was noticed between serum sST2 levels and C-reactive protein (CRP), while there was no significant correlation with serum amyloid protein (SAA) or IL-6. More interestingly, researchers highlighted a negative correlation between CD4+ and CD8+ T lymphocyte counts and sST2. To date, several studies have demonstrated that the COVID-19 patient phenotype is often characterized by low lymphocyte counts and impaired cytotoxic activity [74]. Therefore, researchers hypothesized that high levels of sST2 may reduce IL-33 activity, thus inducing T-cell dysfunction and worse disease outcome. The role of IL-33 among SARS-CoV-2 infections was explored not only in vivo but also through post-mortem studies, which produced notable results. Indeed, Gaurav et al. [75] compared lung sections of COVID-19 patients to those of normal patients (namely, controls) and patients with chronic inflammatory lung diseases including IPF and COPD. They noticed that tissue IL-33 levels were higher among the latter, whereas patients with COVID-19 had very low IL-33 expression, which was significantly reduced as compared with that of control subjects. On the other hand, IL-33 was increased in post-COVID fibrosis, and it was higher compared to the levels in COPD and IPF cases. In conclusion, these data suggested that IL-33 is a key mediator not only in the acute phase of SARS-CoV-2 infection but also in wound repair and lung recovery processes that can be characterized by dysregulated fibrosis. As SARS-COV-2 infection involves the entire respiratory tract, it is worth mentioning the study by Jeican I et al. [76]. Specifically, researchers proposed a comparative morphological characterization of the respiratory nasal mucosa in chronic rhinosinusitis with nasal polyps (CRSwNP) versus COVID-19 and tissue IL-33 levels. Surprisingly, their results aligned with those of Gaurav et al., as IL-33 was lower in COVID-19 patients compared to those with CRSwNP. Certainly IL-33 plays a fundamental role in SARS-CoV-2 infection; however, there are still many aspects to be clarified concerning the variation of this cytokine with regards to disease stage and organ specificity. Furthermore, future studies could be aimed at analyzing IL-33 levels in relation to possible therapeutic agents such as Astegolimab, a human IgG_2_ mAb that selectively inhibits the IL-33 receptor (ClinicalTrials.gov Identifier: NCT04386616). Moreover, another interesting aspect to explore could be the effects of vaccinations on IL-33 levels, thus contributing to a better comprehension of SARS-CoV-2 infection and its impact on the immune system [77].

To sum up, the evidence seems to confirm correlations between IL-33 levels and disease severity, although some results are not completely in line with it. This may be explained by the fact that IL-33 is a pivotal mediator, not only in the acute phase of SARS-CoV-2 infection but also in wound repair and lung recovery, which are often characterized by tissue fibrosis. On the other hand, alternative theories focus on the role of the IL-33 receptor, as sST2 levels may reduce IL-33 activity, thus inducing T-cell dysfunction and worst prognosis. Further research aimed at developing agents targeting IL-33 is certain to shed light on the hidden mechanism of this pleiotropic molecule.

## 9. Conclusions and Future Perspectives

The analysis of the pleiotropic effects of IL-33 on multiple immunological cells (macrophages, mast cells) shows that this alarmin has the capacity to recruit eosinophils, macrophages, and Th2 lymphocytes, especially in chronic inflammatory lung diseases such as asthma, chronic obstructive pulmonary disease, and obstructive sleep apnea (OSA), thus leading to the aggravation of chronic inflammation and the progression of these respiratory diseases [78]. From the discussion in this paper, it is, therefore, possible to deduce how IL-33, in particular, contributes to the development of allergic exacerbations by amplifying type-2 inflammation in the course of the respiratory viral infections under consideration. This alarmin is able to stimulate the development of a powerful proinflammatory cascade at the level of the entire respiratory system, contributing not only to a worsening of the outcome of the disease, but also to bronchial remodeling processes already in childhood, predisposing the subject, therefore, to greater clinical severity of the disease in adulthood (Figure 1). However, from our discussion, it is possible to note how IL-33 plays a double role in the context of respiratory viral infections, in particular as regards influenza viruses A and B, for example, where this alarmin seems to act as a protective factor inducing better viral clearance and promoting tissue repair and lung integrity. The same protective function is also recognized in relation to the ability of IL-33 to favor a better clinical outcome of asthma exacerbations following SARS-CoV-2infections, where IL-33 acts in wound repair and pulmonary recovery and where its inhibition seems to be related to the dysfunction of T lymphocytes and to a worsening of the prognosis. In these contexts, therefore, it would seem that inhibition of IL-33 could have a negative effect on the course of the disease.

Recent studies have highlighted a potential bacterial role of *Staphylococcus aureus* (*S. aureus*) and *Peptostreptococcus magnus* (*P. magnus*) in the exacerbation of the clinical manifestations of respiratory allergic diseases. In fact, these bacterial infections determine, through different mechanisms of cell-mediated interaction, the release of superantigens by T and B cells (SAgs) responsible for a severe course of the disease in patients with chronic inflammation of the airways. In this context, IL-33 also plays a fundamental role, which, induced by superantigens, enhances the release of histamine, angiogenic factors (VEGF-A), and lymphangiogenic factors (VEGF-C) from human lung mast cells (HLMCs), consecutively activating ILC2s and Th2 cells. As a result, the phenomena of eosinophilia, bronchial hyperreactivity, and hyperplasia of goblet cells in the airways occur, with consequent enhancement of bronchial inflammatory manifestations and worsening of the clinical outcome of bronchial asthma [79,80,81].

Currently, the targets used for biological therapies in chronic respiratory diseases include mediators such as IgE and IL-5. Biological drugs in use include omalizumab, the first licensed monoclonal antibody for the management of asthma, which acts by inhibiting the binding of IgE to its receptor expressed on mast cells and basophils, limiting the degree of release of mediators of the allergic response. The monoclonal anti-IgE therapies are accompanied by the use of anti-IL-5 treatments (mepolizumab, reslizumab, and benralizumab) in adults with severe eosinophilic asthma, which is fully endorsed in national and international guidelines [82]. New biological drugs that target innate cytokines and are currently being tested include the human anti-IL-33 mAb molecule (REGN3500), which has shown initial promise in improving asthma control and lung function compared to placebo [83]. Other biological drugs capable of selectively blocking IL-33 are currently in a phase 2a study; among these we find molecules such as GSK3772847 and AMG282 [84,85]. Warren KJ et al. demonstrated that anti-IL-33 antibodies not only lead to a significant reduction in bronchial cytokine and chemokine secretion but are potentially effective in further chronic lung disorders (COPD, IPF), which are often driven by a type-3 inflammatory profile involving excessive levels of neutrophils [86]. In this sense, blocking IL-33 could significantly reduce allergic inflammation, thus representing a promising alternative approach to the treatment of asthma exacerbations following respiratory virus infection.

## Figures and Tables

**Figure 1 biomedicines-10-00715-f001:**
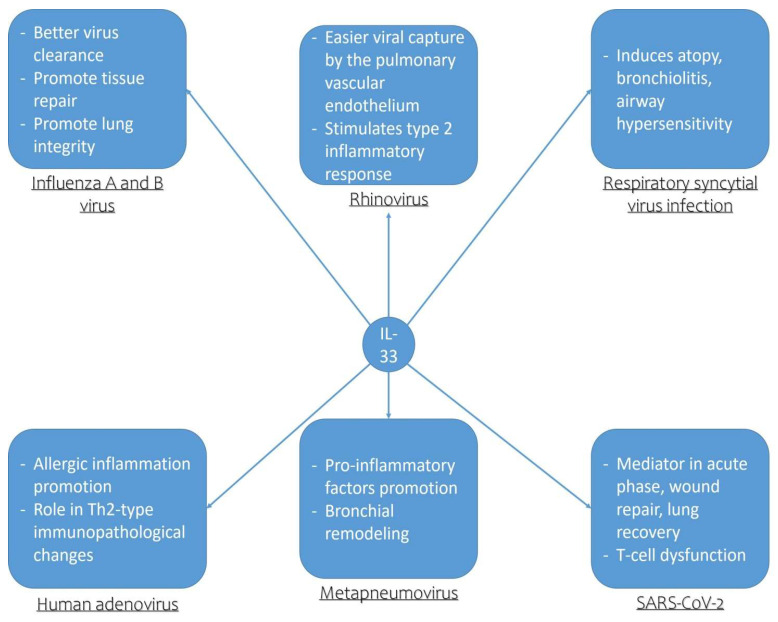
Pathogenetic role of IL-33 in respiratory viral infections.

**Table 1 biomedicines-10-00715-t001:** Research articles analyzed and their main results.

Author	Year	Type of Study	Objective	Outcome
Ronan Le Goffic et al. [11]	2011	Research study	Evaluation of IL-33 expression and release in lungs of influenza A virus-infected mice in vivo and in murine respiratory epithelial cells	Significant increase in mRNA expression of IL-33 in the virus-infected mice at day 3, compared with non-infected control mice. A significant correlation was found between IL-33 mRNA induction and mRNA levels of TNF-a, IFN-g, and IL-6, but not with IL-1b. The protein expression of IL-33 in virus-infected lungs and in BAL was significantly higher than in controls, especially at day 3. We found a significant increase in IL-33 mRNA expression in IAV-infected transformed murine respiratory epithelial cell line, MLE-15, and human pulmonary epithelial cell line A549, compared with non-infected cells.
Jia-Rong Bian et al. [12]	2014	Research study	Evaluation of serum levels of pro-inflammatory cytokines in adult patients with seasonal influenza infection	Higher serum concentration of IL-6, IL-33, and TNF-α at admission, compared to controls. Higher levels of IL-33 were found in influenza A-infected patients.
Kayamuro H et al. [13]	2010	Research study	Evaluation of specific antibody response and specific cellular toxicity in intranasally immunized mice with hemagglutinin and various cytokines	IL-1 family cytokines were able to induce the highest IgG and IgA production. IL-33 and IL-18 were able to elicit both Th1- and Th2-type cytokine responses and high-avidity CD8+ cytotoxic lymphocytes.
Xi-zhi J. Guo et al. [14]	2018	Research study	Evaluation of γδ T cells and production of IL-33 in mouse models of influenza infection	IL-33 induces a local type-2 immune response with increased accumulation of type-2 innate lymphoid cells and T regulatory cells in the lung, which promotes tissue repair and lung integrity after influenza infection.
Monticelli et al. [15]	2011	Research study	Evaluation of the role of innate lymphoid cells after influenza virus infection	Innate lymphoid cells, induced by IL-33, are of crucial importance in promoting airway epithelial integrity and lung tissue homeostasis through production of the epidermal growth factor family member amphiregulin.
Robinson et al. [16]	2018	Research study	Evaluation of the role of IL-33 in mucosal anti-bacterial host defense in influenza infection and bacterial superinfection	Reduction in IL-33 correlates with a negative outcome after influenza infection and bacterial superinfection. Its restoration is correlated with an improvement in bacterial clearance, not related to innate lymphoid cells or type-2 macrophages but to neutrophil recruitment.
Chae Won Kim et al. [17]	2019	Research study	Evaluation of antiviral protection against influenza virus infection by exogenous IL-33	Exogenous administration of IL-33 was related to a better outcome in mice infected with influenza IL-33 virus. IL-33 increased the number of innate lymphoid cells, eosinophils, and dendritic cells and CD8+ T-cell activity.
García-García ML, Calvo C et al. [18].	2017	Experimental study	Investigate whether infants exhibit enhanced nasal airway secretion of TSLP, IL-33, and periostin during natural respiratory viral bronchiolitis	Bronchiolitis caused by common respiratory viruses is associated with elevated nasal levels of TSLP, IL-33, and periostin, factors known to be important in the development of Th2-response.
Mehta AK, Duan W et al. [19]	2016	Experimental study	Examine whether rhinovirus infection of the respiratory tract can block airway tolerance by modulating Treg cells	Infection of the respiratory epithelium with rhinovirus can antagonize tolerance to inhaled antigen through combined induction of TSLP, IL-33, and OX40 ligand, and this can lead to susceptibility to asthmatic lung inflammation.
Jarjour NN, Esnault S [20]	2014	Review	Demonstrate the mechanism of exacerbation in human asthma during rhinovirus infection	IL-33 is likely a major cause of viral-induced asthma exacerbations and a potential therapeutic target in asthma.
Jackson DJ, Makrinioti H et al. [21]	2014	Experimental study	Assess whether rhinovirus induces a type-2 inflammatory response in asthma in vivo and define a role for IL-33 in this pathway	IL-33 and type-2 cytokines are induced during a rhinovirus-induced asthma exacerbation in vivo.
Han M, Rajput C et al. [22]	2017	Experimental study	IL-33 and TSLP expression is also induced by RV infection in immature mice and required for maximum ILC2 expansion and mucous metaplasia	The generation of mucous metaplasia in immature, RV-infected mice involves a complex interplay between the innate cytokines IL-25, IL-33, and TSLP.
Jurak LM, Xi Y et al. [23]	2018	Case-control study	Investigate the effects of IL-33 on rhinovirus (RV)-induced immune responses by circulating leukocytes from people with allergic asthma, and how this response may differ from non-allergic controls	RV infections and IL-33 might interact in asthmatic individuals to exacerbate type-2 immune responses and allergic airway inflammation.
Calvén J, Akbarshahi H et al. [24]	2015	Experimental study	Investigate effects of epithelial-derived media and viral stimuli on IL-33 expression in human BSMCs	RV infection of BSMCs and activation of TLR3 and RIG-I-like receptors cause expression and production of IL-33.
Ramu S, Calvén J et al. [25]	2020	Case-control study	Compare levels of RV-induced IL-33 in BSMCs from healthy and asthmatic subjects	RV infection cause higher levels of IL-33 and increased pro-inflammatory and type-2 cytokine release in BSMCs from patients with non-severe asthma.
Ganesan S, Pham D et al. [26]	2016	Experimental study	Examine the role of TLR2 and IRAK-1 in RV-induced IFN-β, IFN-λ1, and CXCL-10, which require signaling by viral RNA	RV stimulates CXCL-10 expression via the IL-33/ST2 signaling axis, and TLR2 signaling limits RV-induced CXCL-10 via IRAK-1 depletion at least in airway epithelial cells.
Gimenes JA Jr, Srivastava V et al. [27]	2019	Experimental study	Examine the mechanisms underlying the RV-induced persistent inflammation and progression of emphysema in mice with COPD phenotype	RV may stimulate expression of CXCL-10 and IFN-γ via activation of the ST2/IL-33 signaling axis, which in turn promotes accumulation of CD11b+/CD11c+ macrophages and CD8^+^ T cells.
Gajewski A, Gawrysiak M et al. [28]	2019	Experimental study	Analyze the effect of IL-33, the cytokine widely distributed in large amounts in airways of asthmatic individuals, on the HRV-induced inflammatory response in the human lung vascular endothelium.	In asthmatics, IL-33 may facilitate higher viral load in the lung vascular endothelium, while IL-33-orchestrated cytokine milieu may enhance innate inflammatory responses without any concomitant increase in antiviral innate and adaptive mechanisms.
Werder RB, Zhang V et al. [29].	2018	Experimental study	Determine whether anti-IL-33 therapy is effective during disease progression, established disease, or viral exacerbation using a preclinical model of chronic asthma and in vitro human primary airway epithelial cells (AECs)	The latter phenotype was replicated in rhinovirus-infected human AECs, suggesting that anti-IL-33 therapy has the additional benefit of enhancing host defense
Han Xu et al. [30]	2017	Research Study	Evaluation of the role of natural helper cells in influenza virus-induced airway hyper-responsiveness	Blockage of IL-33 reduces natural helper cell recruitment in lungs, thus suggesting IL-33 is necessary for activating Th2-type response.
Wu Yi-Hsiu et al. [7]	2019	Research study	Evaluation of the individual roles of IL-33-activated innate immune cells, including ILC2s and ST2+ myeloid cells, in RSV infection-triggered pathophysiology.	IL-33 is crucial for the activation of ILC2s and the development of airway hyperreactivity and airway inflammation. IL-33 through lung myeloid cells mediates cellular infiltration but not airway hyperreactivity.
Liwen Zhang et al. [31]	2021	Research study	Investigation of the role of NF-κB/IL-33/ST2 axis on RSV-induced acute bronchiolitis	IL-33 level was significantly elevated in infants with RSV acute bronchiolitis. The NF-κB/IL-33/ST2 axis is important in the establishment of the Th2 environment after RSV infection. The use of an anti-IL-33 antibody blocks that mechanism, thus suggesting the crucial role of IL-33, especially produced by macrophages.
Allison E. Norlander and R. Stokes Peebles, Jr. [32]	2020	Review	Review of the impact of L-33, IL-25, thymic stromal lymphopoietin (TSLP), and high mobility group box 1 after RSV infection	ILC2 activation leads to the production of type-2 cytokines and the induction of a type-2 response during RSV infection.
Carolina Augusta Arantes Portugal et al. [6]	2020	Research study	Assess the role of IL-33-ST2 axis in acute lower respiratory infection by RSV	IL-33 and ST2 in nasopharyngeal aspirates on admission were associated with higher risk for mechanical ventilation.
Jing Liu et al. [33]	2015	Research study	Evaluation of IL-13-IL-33-ST2 axis and natural helper cells in the development of RSV-induced airway inflammation	RSV infection induces an increase in the number of IL-13-producing natural helper cells in an IL-33-dependent pathway.
Jordy Saravia et al. [34]	2015	Research Study	Evaluation of the pathogenic mechanisms responsible for RSV-induced immunopathophysiology	Infection with RSV induced rapid IL-33 expression and an increase in ILC2 numbers in the lungs of neonatal mice, in contrast with adult mice. Blocking IL-33 during infection was sufficient to inhibit RSV airway hyperresponsiveness, Th2 inflammation, eosinophilia, and mucus hyperproduction, whereas administration of IL-33 to adult mice during RSV infection was sufficient to induce RSV disease. Elevated IL-33 and IL-13 were observed in nasal aspirates from infants hospitalized with RSV.
Feifei Qi et al. [35]	2015	Research study	Evaluation of cellular source of IL-33, particularly the types of IL-33-producing cells in innate immune cells during RSV infection	IL-33 plays a key role in RSV-induced airway inflammation. Alveolar macrophages and dendritic cells are a cellular source of IL-33. RSV infection increases expression of IL-33 in pulmonary dendritic cells but not in interstitial macrophages. Macrophages and dendritic cells mediate the production of IL-33 through interaction with TLR3 or TLR7.
Feifei Qi et al. [36]	2017	Research study	Evaluation of specific signaling pathways for activation of macrophages during RSV infection	RSV infection can promote both the expression of mRNAs for MAPK molecules and the levels of MAPK proteins in lung macrophages. This mechanism may participate in the process of RSV-induced IL-33 secretion by macrophages, demonstrated by an attenuation of IL-33 production when mice were treated with a special MAPK inhibitor before RSV infection.
Stier MT et al. [37]	2016	Research study	Determination of the capacity of RSV infection to stimulate group 2 innate lymphoid cells (ILC2s) and the associated mechanism in a murine model	RSV-infected IL-33 knock-out mice presented a reduced lung concentration of IL-13, thus highlighting an important role for IL-33 in ILC2 activation.
Zeng S. et al. [38]	2015	Research study	Evaluation and understanding of the function of IL-33/ST2 signaling pathway during respiratory syncytial virus (RSV) infection	Following intranasal infection with RSV, BALB/c mice showed a marked increase in the production of IL-33, with elevated expression of ST2 mRNA as well as a massive infiltration of CD45+ST2+ cells in the lungs, suggesting that during the early phase of RSV infection, IL-33 target cells, which express ST2 on cell surface, may play a critical role for the development of RSV-induced airway inflammation. Indeed, blocking ST2 signaling using anti-ST2 monoclonal antibody diminished not only RSV-induced eosinophil recruitment, but also the amounts of Th2-associated cytokines, particularly IL-13, and Th17-type cytokine IL-17A in the lungs of infected mice.
Bertrand P. et al. [39]	2015	Research study	Evaluation of possible mechanisms that connect RSV bronchiolitis to asthma and recurrent wheezing	Patients with family history of atopy presented a high level of IL-33 in nasopharyngeal aspirates.
García-García ML et al. [18]	2017	Research study	Assessment of the role of thymic stromal lymphopoietin, IL-33, and periostin in viral bronchiolitis	Infants with bronchiolitis had higher levels of TSLP (*p* = 0.02), IL-33 (*p* < 0.001) and periostin (*p* = 0.003) than healthy controls. TSLP and IL-33 were more common in coinfections, mainly RSV and rhinovirus, than in single-infections (*p* < 0.05).
Vu et al. [40]	2019	Research study	Investigate the role of mucosal innate immune responses to RSV and respiratory viral load in infants hospitalized with the natural disease	Levels of IL-4, IL-13, IL-33, and IL-1β were significantly higher in nasal aspirates of patients with severe disease compared with those of patients with moderate disease. The authors highlighted the prevalence of type-2 responses to RSV infection in infants and suggested an important role of ILC2 in shaping the immune response early during RSV infection.
Stav-Noraas TE et al. [41]	2017	Experimental study	Investigate whether endothelial IL-33 expression is augmented by adenoviral activation of the DNA damage machinery	Adenoviral transduction stimulates IL-33 expression in endothelial cells in a way that depends on the DNA-binding protein MRE11 and the antiviral factor IRF1 but not on downstream DNA damage response signaling
Zhang Y et al. [42]	2018	Experimental study	Evaluate the protective effects of Ad5-gsgAM in an ovalbumin (OVA)-induced asthmatic mouse model	Modulating the IL-33/ST2 axis via adenovirus-vectored mycobacterial antigen vaccination may provide clinical benefits in allergic inflammatory airways disease
Yin H et al. [43]	2012	Experimental study	Determine whether high levels of local soluble ST2 can ameliorate ovalbumin (OVA)-induced allergic airway inflammation	Single intranasal delivery of Ad-sST2-Fc to OVA-sensitized mice reduces significantly the production of Th2 cytokines, bronchoalveolar lavage eosinophil infiltrates and histopathological changes in the lung. Moreover, the protective effect is related to blocking IL-33/ST2L signaling.
Stanczak M.A. [44]	2021	Observational study	To analyze the seroprevalence and immune responses in subjects exposed to SARS-CoV-2	− SARS-CoV-2 peptide stimulation elicits IL-33 expression in seropositive individuals. − After recovery from COVID-19, individuals have persisting, circulating PBMCs that produce IL-33 in response to virus-specific T cell activation, which correlates with seropositivity.
Liang Y. [10]	2021	Original article, in vitro study	To test whether SARS- CoV-2 infection induces IL-33 expression in epithelial cells	IL-33 transcript levels significantly increased in cell lines at 72 h post-infection.
Burke H. [45]	2020	Observational study	To measure serum IL-6, IL-8, TNF, IL-1β, GM-CSF, IL-10, IL-33 and IFN-γ using a multiplex cytokine assay, in 100 hospitalized patients with COVID-19	Increased IL-33 levels were associated with adverse outcomes.
Munitz A. [46]	2021	Original article	To investigate a correlation of IL-33 and IgG seroconversion with disease severity	− Notable elevation in the levels of IL-33 was found in moderate/severe patients in comparison with mild patients. − Anti RBD IgG was higher in the moderate/severe cohort, clustered together with IL-33.
Gaurav R. [47]	2021	Observational-post mortem	To characterize IL-33 expression in the lungs of patients with fulminant COVID-19, compared with other inflammatory lung diseases	− Patients with COVID-19 had low IL-33 expression compared with control subjects. − In post-COVID fibrosis, IL-33 was increased compared to levels in COPD and IPF.
Jeican I. [48]	2021	Original article	To perform a comparative morphological characterization of the respiratory nasal mucosa in CRSwNP versus COVID-19 and tissue IL-33	The tissue IL-33 concentration in CRSwNP was higher than in COVID-19.
Markovic S. [49]	2021	Observational study	To analyze the correlation of IL-33 and other innate immunity cytokines with disease severity	− In a more progressive stage of COVID-19, increased IL-33 facilitates lung inflammation. − IL-33 correlates with clinical parameters of COVID-19.
Zeng Z. [9]	2020	Original article	To study the role of soluble ST2 in COVID-19 and its relationship with inflammatory status and disease severity	− Serum sST2 levels were significantly increased in COVID-19 patients and were positively correlated with CRP but negatively correlated with CD4+ and CD8+ T lymphocyte counts. − Serum sST2 levels in non-surviving severe cases were persistently high during disease progression.

## Data Availability

Not applicable.

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
