# Peer review of "Involvement of Il-33 in the Pathogenesis and Prognosis of Major Respiratory Viral Infections: Future Perspectives for Personalized Therapy"

_biomedicines, 2022, doi:10.3390/biomedicines10030715_

Round 1

Reviewer 1 Report

In this review, Murdaca and colleagues present and discuss our understanding of the role of the cytokine IL33 in respiratory virus infections. This appears a well-referenced study, describing much of the literature on IL33 and lung virus infection. Indeed, the table of included studies is much appreciated. Murdaca and all take the approach to focus on separate kinds of viruses and their known interactions with IL33, rather than synthesizing the knowledge together. The authors also include a basic figure, which may be helpful.

General:

I feel that this is a fairly balanced review that focuses on what is known about IL33, rather than remains to be understood. However, there are several gaps in my opinion, the filling of which would help strengthen the review. These are: i) a section on IL33 itself (e.g. production, signalling pathways, downstream induced effectors genes/pathways, the role of nuclear versus extracellular IL33); ii) discussion about why IL33 is protective in some cases (flu) and not in others (asthma); iii) more discussion of the fact that in certain circumstances inhibition of IL33 may have negative consequences for patients; iv) mention of the common bacterial superinfection with viral lung infections and what may be the role of IL33 in those scenarios. In general, this reviewer would have enjoyed a bringing-together of the knowledge in different virus systems and IL33 to identify critical biology. From the current draft it is challenging to do that.

Additionally, I note a few cases of self-referencing of similar reviews from Murdaca et al., in the introduction.

Minor:

Careful with abbreviations - if mentioned once, TLR should be TLR throughout (for example)

Careful with capitalisation of abbreviations e.g. SARS-CoV-2

Define MAPK/JNK etc

Reference needed for intro sentence 3 re: origin of cytokine storm.

What is the "88 gene"?

Paragraphs are long and could be split up more

Some english needed: e.g. sperimental

Author Response

Thaks to the reviewer for his valuable suggestions.

With regard to point i) we have proceeded, as suggested, to add a section dedicated to the pathogenetic and signal transduction role mediated by IL-33.

With regard to point ii) and iii) we have proceeded to insert in the discussion a clarification regarding the dual role of IL-33 in the context of respiratory diseases, which is reported below:

“However, from the discussion that we have carried out, it is possible to note how IL-33 plays a double role in the context of respiratory viral infections, in particular as regards the influenza virus A and B, for example, where this alarmin seems to act as a protective factor inducing a better virus-clearance and promoting tissue repair and lung integrity. The same protective function is also recognized in relation to the ability of IL-33 to favor a better clinical outcome of asthma exacerbations following Sars-CoV2 infections, where IL-33 acts in wound repair and pulmonary recovery and where the its inhibition seems to be related to the dysfunction of T lymphocytes and to a worsening of the prognosis. In these contexts, therefore, it would seem that an inhibition of IL-33 can have a negative effect on the course of the disease.”

With regard to point iv) we have briefly reported the current knowledge on the role of superantigens deriving from respiratory bacterial infections and their link with IL-33 in the pathogenesis of asthmatic exacerbations:

“Recent studies have highlighted a potential bacterial role of Staphylococcus aureus (S. aureus) and Peptostreptococcus magnus (P. magnus) in the exacerbation of the clinical manifestations of respiratory allergic diseases. These bacterial infections, in fact, determine, through different mechanisms of cell-mediated interaction, the release of superantigens by T and B cells (SAgs) responsible for a severe course of the disease in patients with chronic inflammation of the airways. In this context, IL-33 also plays a fundamental role which, induced by superantigens, enhances the release of histamine, angiogenic factors (VEGF-A) and lymphangiogenic factors (VEGF-C) from human lung mast cells (HLMC), consecutively activating ILC2s and cells Th 2. As a result, phenomena of eosinophilia, bronchial hyperreactivity and hyperplasia of goblet cells in the airways occur, with consequent enhancement of bronchial inflammatory manifestations and worsening of the clinical outcome of bronchial asthma [74, 75, 76].”

We have also replaced reference no. 2 ("Murdaca G, Di Gioacchino M, Greco M, Borro M, Paladin F, Petrarca C, Gangemi S. Basophils and Mast Cells in COVID-19 Pathogenesis") as suggested.

With regard to the suggestions of the minor revision, we have corrected and specified what is indicated.  As regards the possible subdivision of the various chapters, we have built our review by including the results about the role of IL-33 in different viral infections, so by further dividing the chapters we think that the continuity of the text would be lost.

Reviewer 2 Report

the study aims to summarize the current knowledge on the role of IL-33  in the pathogenesis and prognosis of major respiratory viral infections. The study is nicely written. There are several formatting issues, such as the table being located at the end of the manuscript. Also, the graphics summarizing the manuscript would be nice. 

Author Response

We thank the reviewer for his valuable suggestions regarding our work. We will fix the formatting problems, as suggested.